# Sociological Importance and Validation of a Questionnaire for the Sustainability of Personal Learning Environments (PLE) in 8th Grade Students of the Biobío Region in Chile

**José Luis Carrasco-Sáez** [1,*] **, Marcelo Careaga Butter** [2,3] **, María Graciela Badilla-Quintana** [2,3] **, Laura Jiménez Pérez** [4] **and Juan Molina Farfán** [5,6]

1 Faculty of Education, Universidad Católica de la Santísima Concepción, Concepción 4090541, Chile
2 Educational Informatics and Knowledge Management Unit, Universidad Católica de la Santísima Concepción, Concepción 4090541, Chile; mcareaga@ucsc.cl (M.C.B.); mgbadilla@ucsc.cl (M.G.B.-Q.)
3 CIEDE-UCSC, Universidad Católica de la Santísima Concepción, Concepción 4090541, Chile
4 ICT Nucleus TIC in Educational and Intercultural Contexts, Universidad Católica de la Santísima Concepción; Concepción 4090541, Chile; Ljimenez@ucsc.cl
5 Educational Informatics and Knowledge Management Unit, Faculty of Education, Universidad Católica de la Santísima, Concepción 4090541, Chile; jmolina@ucsc.cl
6 Faculty of Education, Universidad Católica de la Santísima Concepción, Concepción 4090541, Chile
* Correspondence: jcarrasco@doctoradoedu.ucsc.cl; Tel.: +56-412-345-911

**Abstract:** Contemporary society is going through a cultural transition that leads to new conceptions about the ways in which human beings organize socially and communicate. This process of deep social and cultural transformations is characterized by a technological disruption, in which virtuality forms a new dimension that behaves as an extension of human intelligence. This new form of human interaction impacts on the social imagination, demanding one to rethink social and educational paradigms for the two-dimensional citizen. In this context, this research article describes the sociological importance and the process of social adaptation of users to a personal learning environment (PLE). It includes the validation process of an instrument for the study of the PLE of 8th grade students belonging to 15 schools in the Biobío Region of Chile. A PLE is a frame of reference that can help to understand how two-dimensional citizens socially adapt and influence the sustainability of local and global systems. The validation method for this instrument considered four stages: i) Expert judgment: considering the opinions of six educators and experts in information and communication technologies (ICT); ii) a pilot test: that included a non-probabilistic sample of 472 subjects; iii) a principal components analysis (PCA); and iv) a confirmatory factor analysis (CFA). The Questionnaire on Work Habits and Learning for Professional Futures and the Context Questionnaire SIMCE TIC were used as a reference. When performing a psychometric analysis, a Cronbach alpha coefficient of 0.89 was obtained. This confirms that the adaptation of the instrument is good. The results of the dimensional analysis help us define a structure for the new instrument considering three components that explain 55% of the total variance. The results of the confirmatory factor analysis showed adjustment indexes that support the theoretical model proposed for the PLE study. In conclusion, the instrument was composed of three latent variables: Open self-regulated learning (OSRL) with eight questions, information management (IM) with four questions, and creation and transfer of knowledge (CTK) with four questions.

**Keywords:** validation; questionnaire; personal learning environment; sociology of education

## 1. Introduction

We are living through a cultural transition characterized by the emergence of information and communication technologies (ICT). There is a social importance related to the role of technologies in the behavior of citizens. The sociological change is important when there are two-dimensional performances, i.e., performed in physical spaces and through digital interactions [1–7]. The trends of social change are conformed to the technological disruption, because "the technologies and the user environment are constructed in the same process" [8].

Technologies such as data mining, artificial intelligence, and mass open online courses (MOOCs) are affecting the way people understand knowledge [9].

The concept of a two-dimensional citizen refers to the double face-to-face and virtual dimension of postmodern subjects. They are exposed to the demands of technological disruption. Postmodern citizens solve problems by using the time and space categories of modernity, and at the same time, they communicate, relate, manage information, and manage knowledge in virtual environments. This implies that they develop a unique cultural identity as regards the human group to which they belong, as well as a global identity, at the level of culture on a human scale, which leads them to relate to other people or institutions in real spaces and virtual spaces. What is involved is not only the communicational aspect, but is a profound change of identity.

The functioning of institutions and interpersonal relationships is also changing. Two-dimensional citizens are projected in the near future as a phenomenon of technological disruption. They live in intelligent cities and communicate on a global level. As this social trend grows, new needs, associated with the new individual and his/her organizational performance, emerge. However, we observe that the education system is not explicitly incorporating these changes into the curriculum. There is a tendency to teach confined to the epistemological limits of time and space, these being the modern dimensions of knowledge. This shows that the new complexities of learning are still not identified and the difference between being a consumer of information and being a knowledge manager cannot yet be recognized. In these dynamics, unprecedented in human history, the two-dimensional identity of the citizen is configured, which simultaneously performs in the space of places and in the space of information flows [2], without necessarily being able to become a self-regulated manager of their apprenticeships, maintaining very high levels of intellectual dependence on those who teach them. It is necessary to rethink society and education, in order to reconceptualize emerging models of social, educational, and cultural sustainable behavior so that they allow for the analysis and systematization of these two-dimensional realities.

This scenario of transition, which affects the cultural singularities of human groups as they necessarily have to be linked to a global dimension of culture on a human scale, requires a new pyramid of needs as a frame of reference to help understand how the human being, in its new bi-dimensional condition, can contribute to the sustainable development of its local and global environment. This requires a new sociological conceptualization, since cultural processes such as transculturation, acculturation, and endoculturation have become more dynamic. We observe that transculturation shows a more intense communicational contact between human groups than in previous historical periods, transferring cultural features from a singular human group to other cultural groups. In the acculturation process, there is a tendency towards the loss of cultural identity elements and with regards to endoculturation we see an attempt to reaffirm the authentic identity of each cultural group [10].

Bronfenbrenner [11] states that we move within different systems: The microsystem, mesosystem, exosystem, and macrosystem. A two-dimensional citizen, mediated by technologies throughout his/her personal learning environment (PLE), could adapt to the conditions of each system (physical space and virtual space) and extend his/her influence to the social environment with which he interacts, as shown in Figure 1.

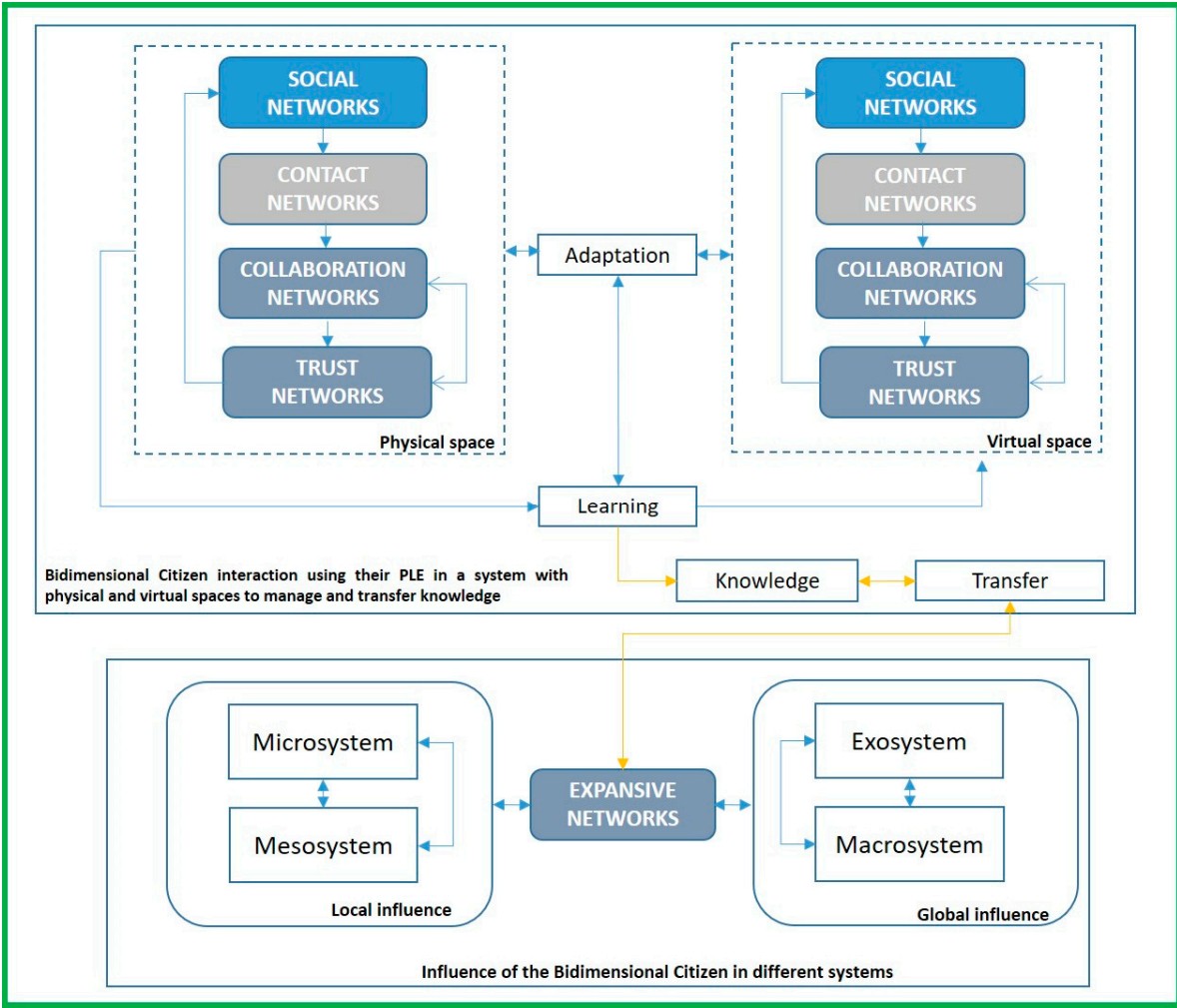

**Figure 1.** Representation of the two-dimensional citizen in interaction with his/her personal learning environment (PLE) (Source: Prepared by the authors based on Bronfenbrenner [11] and the 5R Model [12].

The interesting thing about this representation is that the educational system remains only as an additional system of learning experiences. The school ceases to be the exclusive center of learning, since in the mixed reality (real + virtual) the different systems complement each other to generate learning.

This context of change provides us with new ways of thinking about society, education, the way in which information is accessed, and how knowledge is created and transferred [9,13–17]. A concept that has proliferated in international research and helps us to address this challenge is the personal learning environment.

The PLE is more of a strategy or approach than a particular tool. It is created by the individuals themselves, promoting their autonomous and interconnected learning in a digital ecosystem composed of means, tools, and services [18]. In this way, open, formal and informal learning is promoted, decentralized from the rigidity of the training institutions [19]. This PLE can be defined as the "set of tools, sources of information, connections and activities that a person uses assiduously to learn" [20]. Its basic structure is made up of three elements associated with three cognitive processes: tools and reading strategies (to access information sources), tools and strategies for reflection (environments or services where information can be transformed), tools and relationship strategies (environments to interact with other people from whom it is possible to learn), as stated by Attwell [21].

The importance and international relevance of the PLE has converged in a large number of research projects. Gallego-Arrufat and Chaves-Barboza [22] conducted a review of empirical research

published since 2009, concluding that some studies are grouped around the theoretical and pedagogical justification of the PLE [22–27]. They state that around the concept it is possible to find two lines of significance that are useful for categorizing empirical research. A limited one, which deals with the technological tools of the PLE in order to access new sources of knowledge [28–30], and a wide one, which promotes the PLE as a great economic, political, physiological, intellectual, social, physical, and virtual toolbox for learning [31,32].

In summary, research tends to emphasize that the PLE puts the individuals as the leading characters of their own learning process in bi-dimensional contexts [22,24,33], enabling them to assume new roles, setting their own goals, choosing and organizing their content and technological tools, interacting with friends, family, or teachers, and reflecting on their learning objectives [34–38] regardless of geographic location [23,25]. However, the integration of technology does not necessarily ensure a greater control over learning [34], nor over the appropriation of part of the individual about his/her role as a citizen, whose actions impact on him, in his/her local and global environment.

More empirical research is needed that associates the PLE with the improvement of learning [22] in two-dimensional contexts; and with its contribution as a systemic framework of mobility between the two-dimensional citizen and the diverse systems that influence his/her way of acting before the world.

Despite the international evidence, in Chile there is little research on the use of the PLE in both primary and secondary education or higher education. The only empirical reference found so far was a study carried out by the Ministry of Education in the SIMCE TIC (National Measurement System of ICT Competences) [39] performed in 2013 involving 11,185 second-grade students, belonging to 492 state-run, semi-subsidized, and private schools. The results revealed that only 1.8% of these demonstrated an advanced level of performance in the use of ICT for learning in four dimensions (information, communication, ethics and social impact, functional use of ICT).

In this context, the main objective of this paper is to validate an instrument to determinate the dimensions that should be included in a PLE oriented to students and their environment, in an effort to contribute to the contextualization of the educational challenges of the two-dimensional citizen and his/her influence in different physical and virtual environments.

## 2. Materials and Methods

### 2.1. Research Design

The research was carried out through a non-experimental, descriptive, and transactional study [40,41] during the second semester of the academic year in 2017. The researchers did not have direct control over the analyzed variables that constituted the first version of the instrument.

### 2.2. Participants

The selection of subjects involved in the study was made through a non-probabilistic convenience sampling [42]. A non-random sample composed for 472 Chilean 8th grade volunteer students from both primary state-run and private schools was used. The average age of the students who participated in the study was 13 years. The schools, which are located in Concepción, previously authorized the application of the instrument. This was applied to all 8th grade students who were present.

Regarding the size of the sample, there are two positions as regards carrying out a principal components analysis (PCA), according to Lloret-Segura, Ferreres-Traver, Hernández-Baeza, and Tomás-Marco [43]: i) Those that suggest a minimum size (N); and ii) those that define a proportion of people by item (N/p). For the selection of the sample, the recommendations of the authors that suggest a proportion of people per item were followed. These can vary between 5 to 15 subjects per question, with a size of no less than 100 people [44–46].

### 2.3. The Questionnaire

The first version of this instrument was developed taking into consideration two theoretical backgrounds: The Questionnaire on Work Habits and Learning for Professional Futures and the SIMCE TIC. The first one was developed in the context of the CAPPLE project (Competencies for Lifelong Learning based on the use of PLEs), which seeks to study how the personal learning environments of future Spanish university professionals will be [47]. The author carried out the validation of this instrument through three methods: An expert judgment, a round of cognitive interviews, and a pilot test. The final version of this instrument was made up of 48 questions and four dimensions: Self-perception (alpha = 0.863); information management (alpha = 0.946); management of the learning process (alpha = 0.885); and communication (alpha = 0.772).

The second theoretical background was the SIMCE TIC, which is a national assessment of ICT skills for learning applied by the Ministry of Education in Chile [39]. Once the instrument was constructed, the language was adapted and the questions were regrouped.

The first version of the instrument was made up of five dimensions, fourteen categories, and thirty questions. Twenty questions, two dimensions, and eleven categories came from the Questionnaire on Work Habits and Learning for Professional Futures and twelve questions were adapted from the SIMCE TIC questionnaire. Five new questions were added: Q24, Q28, Q13, Q19, and Q23; three dimensions were added: Access information, manage information, and create knowledge; and also three categories were added: Connectivity, digital identity, and transfer (see Table 1).

**Table 1.** Dimensions, categories, and questions of the first version of the Questionnaire for the study of personal learning environments.

| Dimension | Category | Questions |
|---|---|---|
| Access to information | Connectivity | Q24, Q27, Q28 |
| Information management | Information search | Q4, Q8, Q18 |
| | Organization of the information | Q1, Q11, Q14 |
| Create knowledge | Information processing | Q9 |
| | Ethical processing of the information | Q12 |
| | Knowledge creation | Q6, Q15 |
| Communicate | Team work | Q7 |
| | Digital identity | Q13 |
| | Knowledge transfer | Q5, Q19 |
| Management of the learning process | Intrinsic motivation | Q2, Q16 |
| | Critical thinking | Q10, Q21 |
| | Open learning | Q23, Q25, Q26, Q29, Q30 |
| | Problem resolution | Q22 |
| | Regulation and planning of learning | Q3, Q17, Q20 |

The operationalization of the variables to be evaluated in dimensions and indicators of the variable serves to determine with precision the points on which to obtain information through the corresponding items of the questionnaire. The first dimension access to information corresponds to the devices and strategies that a person uses to access information sources. The second dimension corresponds to manage information, which refers to the devices and strategies that a person uses to find and organize information. The third dimension is to create knowledge, which is associated with the devices and strategies that a person uses to process data and transfer information. The fourth dimension is to communicate, related to the devices and strategies a person uses to critically analyze information, learn in different contexts, solve problems, and plan their learning; and the fifth dimension, management of the learning processes, refers to the devices and strategies a person possesses for team working, managing his/her digital identity and transferring knowledge.

2.3.1. Validation of the Instrument

The validation method considered four processes: i) The validation of the construct through expert judgment, ii) the pilot test: that included a non-probabilistic sample of 472 subjects; iii) the principal components analysis; and iv) the confirmatory factor analysis (CFA).

*Process 1. Expert Judgement*

According to Crocker and Algina [48], expert judgment allows the validation of a research instrument, being a useful procedure to realize its content validity [49]. As a result of the difficulty of having all the experts at the same time, expert judgment was performed using the method of individual aggregates. A group of six experts in education and information technology was selected in Chile. Each one received the validation format of the instrument, consisting of a table for each question, identifying the dimension to which they belonged. These experts independently assessed the importance of each questions, according to three criteria: i) Uniqueness of language, to validate that the question was clearly written and does not lead to more than one interpretation, answering Yes or No; ii) relevance of the question with respect to the dimension in which it has been raised, answering Yes or No; iii) importance of each one of the items of the questionnaire, answering a Lickert scale from 1 to 5, where 1 was the Minimum importance, and 5 the Maximum importance. In addition, there was a space for observations so that the evaluators could write the comments they deemed pertinent.

On the basis of the suggestions and comments, nine questions were eliminated from it: Q2, Q3, Q5, Q8, Q11, Q12, Q15, Q18, and Q21, the wording of some items was modified, and some of the questions were regrouped (see Table 2).

**Table 2.** Regrouped questions based on expert judgment.

| Regrouped Questions | New Questions |
|---|---|
| Q2 - Q3 | Q2 |
| Q5-Q8-Q15 | Q11 |
| Q11-Q12-18 | Q5 |
| Q21 | Q10 |

The problem solving category was eliminated, and the information processing and ethical processing of information categories were merged into a new category called information processing. The new version of the instrument, used as a pilot test, was finally composed of 5 dimensions, 21 questions, distributed in 12 categories as can be seen in Table 3.

**Table 3.** Dimensions, categories, and questions of the pilot version of the instrument.

| Dimension | Category | Questions |
|---|---|---|
| Access to information | Connectivity | Q14, Q19, Q20 |
| Manage information | Information search | Q2 |
| | Organization of the information | Q1 |
| Create knowledge | Information processing | Q5, Q8 |
| | Knowledge creation | Q3 |
| | Team work | Q4 |
| Communicate | Digital identity | Q7 |
| | Knowledge transfer | Q11 |
| | Intrinsic motivation | Q9 |
| Management of the learning process | Critical thinking | Q6 |
| | Open learning | Q15, Q16, Q17, 18, Q21 |
| | Regulation and planning of learning | Q10, Q12, Q13 |

*Process 2. Pilot Test*

A pilot test was carried out with 472 8th graders, belonging to state-run, subsidized, and private schools in Concepción, Chile. The participants were those students who offered their consent as well

as their parents' or tutors'. In that way the confidentiality of the information was guaranteed. The data collection was carried out during the second semester in 2017 and it was applied as a printed document. A brief explanation of the questionnaire and the process was given to the students before they started answering. An example of the questions was provided in order to help them understand how they should answer the questionnaire. The application time was approximately forty-five minutes.

### 2.3.2. Statistic Analysis

To carry out the evaluation of the elements that make up the instrument, internal consistency measures were calculated using the Cronbach's alpha coefficient and correlation, through the Pearson correlation coefficient. In order to determine if a PCA was possible, the Barlett's sphericity test and the sample adequacy measure (SAM) were used. The principal components analysis was performed and an oblique rotation using the Oblimin technique. The results obtained in this test were only applied to the eighteen questions that had variables of scalar type. For the confirmatory factor analysis, the maximum likelihood (ML) method was used and five indices were considered: $X^2$ (Chi square), RMSEA (mean square error of approximation), CFI (comparative adjustment index), TLI (Tucker Lewis index), and SRMR (root mean square residual standardization). The analyses were performed with the statistical software R, version 3.5.1.

## 3. Results

### 3.1. Principal Components Analysis

The first exploration of the data shows that some students did not answer all the questions and they left some empty items (without answering). Therefore, the valid sample consists of 439 subjects, as shown in Table 4. Regarding sex, it could be observed that the proportions are very similar between boys (49%) and girls (51%).

**Table 4.** Valid sample of participants.

| Type of School | Boys | Girls | Total |
| --- | --- | --- | --- |
| Private and state-subsidized (PSS) | 50 | 48 | 98 |
| State-subsidized (SS) | 166 | 175 | 341 |
| TOTAL | 216 | 223 | 439 |

The answers were analyzed according to a Lickert scale from 1 to 7, where 1 was Totally Disagree, and 7 the Totally Agree (Table 5).

**Table 5.** Types of answers.

| Meaning | Abbreviation | Answer |
| --- | --- | --- |
| Totally Disagree | TD | 1 |
| Strongly Disagree | SD | 2 |
| Disagree | D | 3 |
| Undecided | UD | 4 |
| Agree | A | 5 |
| Strongly Agree | SA | 6 |
| Totally Agree | TA | 7 |

The reliability analysis of the instrument yielded a Cronbach's alpha coefficient =0.89. A Pearson correlation matrix was used to analyze the existing collinearity between the variables. A $p < 0.01$ value was obtained when applying the determinant of the matrix. This suggests a high level of collinearity in the set of variables of the correlation matrix. For the multicollinearity assumptions, the Bartlett sphericity test was used, which yielded a $p$ value of $< 0.0001$; and the sample adequacy measure

Kaiser–Meyer–Olkin (KMO) test yielded a value of 0.92. According to the results obtained, a principal components analysis was carried out. In order to extract the components, we analyzed two criteria: The parallel analysis (using the function fa.parallel, in the R Studio software), which yielded the suggestion of three components and then we complemented it with the variance percentage criterion that refers to the principal components analysis, taking the variance explained by the three components suggested in the previous analysis. The three components explained 55% of the total variance. Table 6 summarizes the results of the first rotation, using oblique rotation.

**Table 6.** Matrix of rotated components for the first analysis.

| Questions | Components | | |
|---|---|---|---|
| | 1 | 2 | 3 |
| Item_17 | 0.83 | | |
| Item_16 | 0.82 | | |
| Item_18 | 0.75 | | |
| Item_15 | 0.68 | | |
| Item_14 | 0.58 | | |
| Item_11 | 0.51 | 0.43 | |
| Item_9 | 0.50 | | 0.32 |
| Item_12 | 0.50 | | |
| Item_13 | 0.36 | 0.33 | |
| Item_4 | | 0.86 | |
| Item_3 | | 0.81 | |
| Item_5 | | 0.56 | |
| Item_6 | | 0.46 | |
| Item_2 | | 0.40 | 0.34 |
| Item_7 | | | 0.86 |
| Item_10 | | | 0.51 |
| Item_8 | | | 0.50 |
| Item_1 | | | 0.45 |

Items 2 and 13 scored similarly in more than one dimension, influencing the total variance explained. It was decided to carry out a new rotation of components excluding the mentioned items. The result is presented in Table 7.

**Table 7.** Matrix of rotated components for the second analysis.

| Questions | Components | | |
|---|---|---|---|
| | 1 | 2 | 3 |
| Item_17 | 0.83 | | |
| Item_16 | 0.81 | | |
| Item_18 | 0.75 | | |
| Item_15 | 0.68 | | |
| Item_14 | 0.57 | | |
| Item_9 | 0.50 | | 0.32 |
| Item_11 | 0.49 | 0.39 | |
| Item_12 | 0.49 | | |
| Item_4 | | 0.84 | |
| Item_3 | | 0.80 | |
| Item_5 | | 0.59 | |
| Item_6 | | 0.47 | |
| Item_7 | | | 0.86 |
| Item_10 | | | 0.52 |
| Item_8Item_1 | | | 0.500.46 |

On the basis of the results obtained in the validation process, the final version of the instrument was modified and two questions (Q2 and Q13), three dimensions (access to information, communication, management of the learning process), and two categories (connectivity, critical thinking) were eliminated.

The other categories were regrouped into three dimensions. The 16 items left were distributed into three components as follows: Component 1 called "open self-regulate learning" (OSRL) with eight questions; Component 2 called "information management" (IM), with four questions; and Component 3, identified as "creation and transfer of knowledge" (CTK), with four questions. The OSRL dimension refers to the devices and strategies that motivate a person to learn and transfer knowledge in bi-dimensional contexts.

The analysis of the Cronbach's alpha of each of these components yielded an alpha =0.87 for the OSRL component; an alpha =0.86 for the IM component; and an alpha =0.70 for the CTK component. Out of the 19 questions available in the final version of the questionnaire, 16 correspond to a scale of seven levels, one corresponds to frequency, one to Yes/No and I do not know answer; and a question with answers of the type I learned: alone/with my teachers/with my family/with my friends/I do not know. The final structure composed of three validated dimensions, which explain 55% of the total variance is presented in Table 8.

**Table 8.** Components, categories, and questions (final version of the instrument).

| Dimension | Category | Questions |
|---|---|---|
| Open self-regulated learning | Open learning | Q12, Q14, Q15, Q16 |
| | Regulation and planning of learning | Q11, Q13 |
| | Intrinsic motivation | Q8 |
| | Transfer knowledge | Q10 |
| Information management | Search for information | Q4, Q5 |
| | Information processing | Q2 |
| | Teamwork | Q3 |
| Creation and transfer of knowledge | Knowledge creation | Q7 |
| | Organization of information | Q1 |
| | Digital identity | Q6 |
| | Feedback | Q9 |

Regarding the correlations between the components, a weak positive correlation was found between OSRL and IM ($r_S$ = 0.42, $p$ <0.001); between OSRL and CTK ($r_S$ = 0.48, $p$ <0.001) and a moderate positive correlation between IM and CTK ($r_S$ = 0.51, $p$ < 0.001).

*3.2. Confirmatory Factor Analysis*

To corroborate the results of the psychometric properties obtained in the principal components analysis, a confirmatory factor analysis was carried out, which allows one to represent the relations of the latent variables with their observed or indicator variables [50,51], and confirm that all the questions fit the proposed model [52–54].

The analyzed model consists of three components: Open self-regulated learning (OSRL), information management (IM), creation and transfer of knowledge (CTK) that influence a group of observed variables, measured through the questions of a scale [55].

Using the Lavaan package [56], from the statistical software R, we can represent the model to be analyzed in the following way:

$$\text{Model} <-\text{'} OSRL = ~\sim Q08 + Q10 + Q11 + Q12 + Q13 + Q14 + Q15 + Q16$$
$$IM = \sim Q02 + Q03 + Q04 + Q05$$
$$CTK = \sim Q01 + Q06 + Q07 + Q09$$
$$OSRL \sim\sim IM + CTK$$
$$IM \sim\sim CTK\text{'}$$

where each command is explained as follows:

= ~ allows for the inclusion of regression relations that define a latent variable; ~ relationship between two observable variables; and ~~ that specifies the relationships of variance and covariance [51].

To evaluate the goodness of fit of the model, the maximum likelihood (ML) method was used and five indices were considered (Table 9). The $X^2$ statistic (square Chi), which allows us to identify the best possible fit between the compared matrices [57]. A value of 342.377 ($p = 0.000$) was obtained, with degrees of freedom (DF) =101. Although the value of p is less than 0.05 (the model and the data do not fit each other), it is suggested not to consider this statistic because of its sensitivity when used in samples with more than two hundred subjects [51,58–60].

**Table 9.** Indexes of goodness of fit of the model.

| Indexes of Goodness of Fit of the Model | $X^2$ | Df | p | CFI | TLI | RMSEA | SRMR |
|---|---|---|---|---|---|---|---|
| Model | 342.377 | 101 | 0.000 | 0.92 | 0.91 | 0.07 | 0.04 |

The other indicators used are less sensitive to differences in the sample size. The mean square error of approximation (RMSEA) obtained was = 0.07, considered to be within the acceptable range as an adequate adjustment of the model [51,61–63]. The comparative adjustment index (CFI), which compares the proposed model with a null alternative model [62,64], was 0.91, which means that the model for measuring the questionnaire and the structure of the data reproduces at least 90% of the covariance, considered as the minimum acceptable value [51,63,65]. The SRMR index (standardized version of the RMSR index) obtained a value of 0.04, indicating a good adjustment of the model [51,55]. Finally, the non-regulated adjustment inside (TLI) was considered, whose value was 0.91, considered as a good adjustment of the model [66].

## 4. Conclusions

The incorporation of the virtual dimension to human activities is causing a cultural transition, characterized by a new generation of citizens that develop both in physical and virtual environments; these were called 'space of the flows' by Castells [2] or the 'resonant interval' by McLuhan and Powers [3].

The digital citizens of the cultural transition are exposed to profound changes in their human behavior. These tendencies are associated with the technological disruption that we experience and which characterize the so-called fourth industrial revolution, provoking new needs and demanding redefinitions of roles and new scenarios in education. In the near future, the new two-dimensional citizens will overcome the limits of time and space of the traditional modern classroom, actively incorporating cyberspace into more autonomous and self-regulated learning processes.

It is a complex redefinition of a new subject that in its identity needs to be analyzed through three important conceptions: A new human condition; how to access, create, and share knowledge; and the recognition of the new essence of an individual with emerging needs.

During the last century, Maslow [67] was able to recognise the needs of the subject as an individual and its relationship with society, but from a vision located in the modern categories of analysis (time and space). However, new needs have been incorporated, based on a mixed reality that involves the real world and the virtual world, which come from the process of a growing technologization of people, society, and culture. To the extent that a greater number of subjects, understood from the technological perspective as excessively complex and probabilistic natural systems, are related to other people or institutions using automated means of communication, these new communicative interactions are produced in complex patterns of decisions and control for achievement of the purposes. These processes, typical of postmodernity, require new skills and the growing satisfaction of new needs to shape the profile of a two-dimensional citizen. Their levels of complexity and influence increase from the subjects, from the local culture towards the global culture.

These questions pose challenges for education in this cultural transition. It is possible to confirm, according to different authors, that the formal education system continues to be decontextualized before the needs of the subjects in two-dimensional contexts [6,7,9,17]. Incompatibilities arise between the society that learns with the new technologies and the one that learns exclusively with the traditional methods of a school centered in the teacher, in standardized evaluations, and knowledge located only within the limits of the classroom [9,18], without addressing the specific needs of the cultural transition.

In this context, this study describes the process of social adaptation of two-dimensional citizens and the validation of an instrument used with 8th grade students in the Biobío region of Chile. This work will contribute to the study of PLE in similar contexts. A PLE is a frame of reference that can help to understand how two-dimensional citizens adapt in their social interactions and influence the sustainability of local and global systems. Knowing more about students' PLE will allow teachers in the education system to design training routes that prepare them with knowledge, skills, attitudes, and values to contribute to a vision of sustainable global development [10].

It also provides an instrument to build a reconceptualization of educational sociology, by measuring the new roles of students, preferably engaging with the concepts of open learning, information management, and the creation and transfer of knowledge. [8,68].

A final version made up of 18 questions was obtained based on the work of Prendez-Espinoza et al. [47] and the SIMCE TIC context questionnaire [39]. The questions were grouped into three main components (OSRL, IM, CKT) that explain 55% of the total variance. The validation process included expert judgment, a pilot sample, a principal components analysis, and a confirmatory factor analysis. For the principal components analysis, two psychometric tests were performed: The Bartlett sphericity test (to ensure that the correlation matrix is not the same as the inverse or identity matrix), which yielded a value $p < 0.0001$ (significant); and the measure of sample adequacy, known as the Kaiser–Meyer–Olkin test (KMO), whose value was 0.92. The total alpha of the instrument was =0.89. To evaluate the fit of the proposed model, five goodness-of-fit indices were used: $X^2$ (Chi square), RMSEA, CFI, TLI, and SRMR. The results were favorable, since it was possible to obtain four acceptable values in the indices described, except in the case of $X^2$.

Despite the international relevance of the PLE concept, there is little research in Chile and Latin America [68,69] that studies in depth the characteristics and students' perception of their personal learning environments. With this article we hope to contribute with an instrument that can be used by other researchers in different contexts in order to validate the final components associated with PLE described in this incipient research line and to examine in detail the results obtained in this study.

Finally, the authors suggest the final validated version of the instrument that could be used in different contexts and with different groups of people that need to be analyzed in their bi-dimensional interaction.

## 5. Limitations

It is possible that carrying out a PCA and a CFA could eventually lead to an over-adaptation of the factors [70,71]. Future studies should try to replicate the factor structure found. Likewise, the

possibility of collecting new data will allow establishing a nomological network in order to test the new scale with theoretically related constructs, such as success in learning.

**Author Contributions:** J.L.C.-S and M.C.B conceived the original idea of this research. All authors contributed to the writing of this manuscript. L.J.P supported the review of the writing and final version of data analysis. M.G.B.-Q contributed to the analysis of the conclusions and J.M.F reviewed both the Spanish and the English version of the document. All authors have read and approved the final version of the manuscript.

**Funding:** This research received no external funding.

**Acknowledgments:** This work has been written thanks to the support of the National Commission of Scientific and Technological Research, CONICYT, Ministry of Education, Chile, through the Postdoctoral Scholarship Abroad, BECAS CHILE, granted to María Graciela Badilla Quintana [Project No. 74160087] and Doctoral Scholarship in Chile (National Doctorate Scholarship), BECAS CHILE, granted to José Luis Carrasco-Sáez [No. 21171389]. We are grateful for the support of the CIEDE-UCSC Center for Research in Education and Development.

**Conflicts of Interest:** The authors declare no conflicts of interest.

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
