# Peer review of "Sociological Importance and Validation of a Questionnaire for the Sustainability of Personal Learning Environments (PLE) in 8th Grade Students of the Biobío Region in Chile"

_sustainability, doi:10.3390/su11051301_

Round 1
Reviewer 1 Report
This article reports the process of adapting and validating an instrument for studying the dimensions of Personal Learning Environments (PLE). This is an interesting topic and the paper as a whole would be of interest to the journal audience. However, major changes should be undertaken for the article to be considered for publication in Sustainability.
The introduction does not provide sufficient background to justify the relevance and the need for validating the instrument. The rationale is slightly confusing as it refers to concepts and arguments that do not seem relevant for the purpose of the study (lines 40 – 59). Relevant information regarding PLE is presented, however many of the references and the literature does not come from high referred journals; instead several references are from website and non-indexed documents. This should be revised in depth. Concepts such us “bi-dimensional citizen” are not defined, and the connection to Bronfenbrenner ecological systems theory is not clearly justified.
Overall, the methods used for the adaptation and the validation of the instrument are clearly justified and reported. However, the selection of the participants should be better explained. Information provided refers only to “a non-random sample” but no details for selecting the students is provided. Another major concern refers to the information regarding the dimensions and items of the instruments used for the adaptation and validation described. On the other hand, the dimensions should be defined, and some items should be included to support the study. Explicit information regarding the items of the instrument or the questionnaire itself should be provided.
The conclusions are rather too short and lack of critical insights from the authors. The scientific significance of the paper should be discussed.
A better link to the idea of sustainability is required. It is not enough to include the word in the title, but it would be interesting to see how the contribution link to the aims and scope of the journal and the particular special issue Special Issue "The Importance of Sociology of Education for a Sustainable Future"
Finally, the writing should be revised and proof reading and overall, the English should be checked. Some minor mistakes are in the text (see abstract line 35 – the word Transfer is repeated).
Author Response
Dear Reviewer
On behalf of the research team, I would like to thank all of your suggestions.
In the attached document you will find the answers to your comments.
Best regards,
José Luis Carrasco-Sáez

Reviewer 2 Report
As PLE increases the students’ motivation, enables building autonomous learner and allows to customise the learning process according to the needs, preferences and abilities of the learner I consider the presented topic to be very important.
Learners in both, formal and informal education are exposed to different types of knowledge and stimuli and they process it differently. Not only digital natives use technologies.
The title of the article suggests that the text is also about the sustainability of PLE and in this sense I would expect to find more about the sustainability of PLE (even though there are some ideas in Conclusions).
The authors mention that they “did not have direct control over the analyzed variables that constituted the instrument’ first version“. Could they be more specific?
In the Research design they mention „the second semester of the academic year in 2017“ and in the Participants description they speak about 8th grade volunteers what might be puzzling (I suggest using months rather than mixing university terminology and terminology used with primary or secondary education) - it might be cultural issue, authirs may or need not accept the suggestion(s).
It might be also useful to mention age of participants as school systems vary in different countries.
The dimensions’ approval is defined (lines 147-155), however, adding reason(s) why authors believe it was necessary to add 5 questions (line 131) (and what questions?)categorised in 4 different Categories might be interesting.
I appreciate detailed and precise description of how and which statistical process were applied, with limitations stated.
The authors inform the reader that a final version consists of 18 questions and are grouped into 3 main dimensions but the reader has no idea about the questions itself (I understand, the test might be the subject of sale and thus items not published in the full form).
Author Response
Dear Reviewer
On behalf of the research team, I would like to thank all of your suggestions.
In the attached document you will find the answers to your comments.
Best regards,
José Luis Carrasco Sáez

Reviewer 3 Report
The authors of the present manuscript aim to validate a questionnaire for the assessment of the sustainability of Personal Learning Environments (PLE). They used expert judgments, exploratory and confirmatory factor analysis for item selection and scale evaluation/validation.
As I am not an expert on the topic of Personal Learning Environments, my comments will only refer to the validation process. Unfortunately, the items of the questionnaire were not provided. Thus, it is not possible to evaluate the content validity of the items.
Regarding the test of the factorial structure, there are some weaknesses in the analysis. If the goal is to obtain latent factors, the authors should use a principal factor analysis not a principal component analysis that is, strictly speaking, not a factor analysis method at all (Fabrigar et al., 1999). Furthermore, the authors used orthogonal rotation, however, oblique rotations are superior, especially when one expects the different latent dimensions to be correlated (e.g., Fabrigar et al., 1999). Moreover, the authors do not report which method they used to determine the number of extracted components (e.g., Kaiser criterion, scree plot, parallel analysis, MAP test, CD test etc.; Ruscio & Roche, 2012). The authors also performed a principal component analysis and a confirmatory factor analysis on the same data that might lead to overfitting (e.g., Fokkema & Greiff, 2017). In fact, the authors should use an additional data set in order to assess if the factor structure is replicable.
Moreover, the authors do not report any test of the convergent and discriminant validity. They should try to establish a nomological network in order to strengthen evidence of construct validity (Cronbach & Meehl, 1955). In summary, the authors only report tests about the content validity, the factor structure and the internal consistency that are important psychometric properties. However, they do not report other important psychometric properties (e.g., convergent validity) that are necessary for a comprehensive scale validation.
Minor points:
· Technically, Table 8 does not display a factor structure.
· I don’t think three correlations need a table (Table 9).
References
Cronbach, L. J., & Meehl, P. E. (1955). Construct validity in psychological tests. Psychological bulletin, 52, 281-302.
Fabrigar, L. R., Wegener, D. T., MacCallum, R. C., & Strahan, E. J. (1999). Evaluating the use of exploratory factor analysis in psychological research. Psychological methods, 4(3), 272-299.
Fokkema, M., & Greiff, S. (2017). Editorial. How Performing PCA and CFA on the Same Data Equals Trouble. European Journal of Psychological Assessment, 33, 399-402.
Ruscio, J., & Roche, B. (2012). Determining the number of factors to retain in an exploratory factor analysis using comparison data of known factorial structure. Psychological assessment, 24, 282-292.

Author Response

(The authors gave the same response as above.)

Round 2
Reviewer 1 Report
The manuscript has improved with regards to its previous version. The authors have made an effort to include a sociological discussion that better relates the paper with the Special Issue.
The conclusions have been extended including a theoretical account. However, there are still some inconsistencies that need to be clarified before publication; some of them were already pointed out in my previous review and others come from this second version:
- The definition of two-dimensional citizens, bi-dimensional citizen, bi-dimensional condition, two-dimensional contexts is not clear. One can infer that you refer to technological vs non-tech dimension, but this is not clearly defined in the paper.
- In the abstract and in the conclusion there are these statements “This research article describes the sociological importance and the process of social adaptation of users to a Personal Learning Environment (PLE)” and “In this context, this study describes the process of social adaptation of two-dimensional citizens”. What do the authors mean by “the process of social adaptation”? How this process of social adaptation has been taken in the study? As far a as I understood, the paper reports the adaptation and validation of an instrument for studying the dimensions of Personal Learning Environments (PLE). This is a major concern that should be addressed clearly, otherwise makes the paper confusing.
- The keyword “social impact” does not represent the contribution of the article
- Citing Maslow in the conclusions should be referred and I am not quite sure how Maslow’s theory is relevant in this contribution.
- Still, explicit information regarding the items or the final validated instrument itself is missed. In the last sentence “the authors suggest the final validated version of the instrument that could be used in different contexts and with different groups of people that need to be analysed in their bi-dimensional interaction”. What does “bi-dimensional interaction” mean? This is a big statement with a concept that should be clarified.
Author Response
Dear Reviewer
I thank the research team for all their suggestions.
Below you will find our document with the answers to your comments.
Thank you.
Best regards.

Reviewer 3 Report
The authors improved the manuscript by changing their analyses and providing additional information on the statistical procedure.
However, one point I still think is important is the problem of overfitting. If the authors do not want to collect a new sample to confirm the factor structure in a new sample, they should at least write a paragraph in a limitation section that performing a pca and cfa might have led to factor overfitting (Fokkema & Greiff, 2017) and that future studies should try to replicate the found factor structure. Furthermore, collecting new data rises the opportunity to establish a nomological network and test the new scale with theoretically related constructs (e.g., learning success).
Minor points
· Please provide the reference of the original article that describe the parallel analysis (that is Horn, 1965 see references).
· Please provide also the chi-square value, degrees of freedom and p-value of the cfa model in Table 9.
· There was a misunderstanding regarding the former Table 9. I was suggesting to put the three correlations in the text instead of putting them in an extra table. I think the correlations between the three dimensions are worth reporting.
References
Fokkema, M., & Greiff, S. (2017). Editorial. How Performing PCA and CFA on the Same Data Equals Trouble. European Journal of Psychological Assessment, 33, 399-402. doi: 10.1027/1015-5759/a000460
Horn, J. L. (1965). A rationale and test for the number of factors in factor analysis. Psychometrika, 30, 179–185. doi:10.1007/BF02289447
Author Response

(The authors gave the same response as above.)

Round 3
Reviewer 1 Report
The revised version of the manuscript has addressed all the comments and suggestions and it can be published as it is.
Author Response
Thanks for all your comments and suggestions that have contributed to improve our article.